## RESEARCH ARTICLE

# Protecting larval fish at water intakes: hydraulic and biological evidence for the effectiveness of modern fish-protection screens

Craig Ashley Boys[1,*], Wayne Robinson[2], Katherine E. Doyle[2], Thomas S. Rayner[1,2], Patrick McSweeney[1] and Lee J. Baumgartner[2]

## ABSTRACT

Water intakes entrain large numbers of fish larvae in waterways where drift coincides with large-scale extraction. While modern fish-protection screens can reduce these losses, many are not designed for larvae and were developed or evaluated primarily for juveniles and adults. This study evaluated the effectiveness of Australia's fish screen design criteria (which specify a maximum approach velocity of 0.1 m s$^{-1}$ and slot widths of 2–3 mm) for protecting drifting larval Murray cod (*Maccullochella peelii*). Larvae were tested in a large flume under combinations of approach velocity (0.1 or 0.2 m s$^{-1}$), slot width (2 or 3 mm), and proximity. Entrainment rose sharply with velocity; slot size had a smaller interactive effect. The most protective combination (0.1 m s$^{-1}$ and 2 mm) reduced entrainment by up to 94% relative to unscreened conditions. Three-dimensional flow measurements helped explain how velocity vectors interact to influence larval fate. The results demonstrate that Australia's current standards, although developed for juveniles, can provide strong larval protection when strictly followed, but that even modest departures can sharply increase risk. More broadly, since the criteria tested here are less conservative than those adopted in many other countries, where empirical evidence on larval behaviour does not exist, targeted research could determine whether existing guidelines warrant revision.

KEY WORDS: Fish screens, Larval entrainment, Approach velocity, Screen design criteria, Slot width

## INTRODUCTION

Many freshwater fish species undergo a period of larval drift, where recently hatched individuals are transported downstream by the current (Lechner et al., 2016). While this strategy facilitates dispersal from spawning grounds, sometimes over thousands of kilometres in a short period (Stuart and Sharpe, 2020), it also exposes larvae to instream infrastructure such as dams, regulators, turbines, pumps, and water diversions. These structures can result in high rates of injury, impingement, and entrainment (the unintentional passage of fish into a diversion) (Boys et al., 2016;

Jarvis and Closs, 2019; Marttin and De Graaf, 2002; Navarro et al., 2019).

Modern fish-protection screens aim to reduce these impacts by preventing fish from being drawn into water diversions. Their effectiveness depends on how well the screen design accounts for the size, swimming ability, and behaviour of the target species. As a result, design guidelines, covering aspects such as mesh size and hydraulic conditions, vary across jurisdictions. Countries like Australia, the USA, New Zealand, and the UK apply their own specifications, which differ in stringency based on local species, environmental conditions, and management priorities (Table 1).

Some guidelines adopt a generalist approach to the species and life stages they aim to protect, referring broadly to "all resident fish" or "all life stages" [e.g. SWIWG, 2021; The Eels (England and Wales) Regulations, 2009; USFWS and NOAA, 2014]. Some explicitly mention larvae (e.g. NMFS, 2022), while others appear focused on juvenile or adult stages (e.g. Boys, 2021; NIWA, 2023; Norlund and Bates, 2000) (Table 1). Regardless of their stated focus, much of the recent guideline development or evaluation across Australia, the USA, New Zealand, and the UK has concentrated on older life stages, particularly juveniles and adults (e.g. Boys et al., 2013a; Carter et al., 2023; Jellyman et al., 2023; Peake, 2004; Stocks et al., 2019; Swanson and Young, 1998; Swanson et al., 2004).

There has been evaluation of screen criteria with respect to larvae, although most of the research is from North America in the 1970s and early 1980s, examining the vulnerability of early life stages to entrainment and impingement at power plant intake structures. A comprehensive review of the research of this period (Chow et al., 1981) highlighted the high susceptibility of larvae to entrainment, particularly when intakes are located near spawning and nursery habitats. The review also synthesised findings from early screen trials, including fine-mesh and wedgewire designs, which demonstrated only partial exclusion of larvae, especially when approach velocities exceeded 0.15 m s$^{-1}$ (e.g. Heuer and Tomljanovich, 1978; Zeitoun et al., 1981). More recently, a multi-species laboratory study confirmed that the effectiveness of screens for larval protection varies substantially depending on hydraulic conditions (e.g. approach and sweeping velocities), screen design (e.g. slot width and orientation), and biological traits such as fish size and behaviour (EPRI, 2003). This complexity, and the species- and context-specific nature of larval responses, makes it difficult to generalise how larvae interact with screens. This is especially true when attempting to apply findings across continents, where fish assemblages, screen technologies, and environmental conditions can differ markedly. The EPRI (2003) study ultimately recommended a more targeted, species-specific approach, with a narrower range of design parameters, to better define the relationships between screen criteria and larval protection.

[1]NSW Department of Primary Industries and Regional Development, Port Stephens Fisheries Institute, Taylors Beach, New South Wales 2316, Australia. [2]Gulbali Institute, Charles Sturt University, 386 Elizabeth Mitchell Dr, Thurgoona NSW 2640, Australia.

*Author for correspondence (craig.boys@dpird.nsw.gov.au)

C.A.B., 0000-0002-6434-2937; T.S.R., 0000-0001-9616-1068

**Table 1. Key design criteria for fish-protection screens set by global regulations and guidelines**

| Location | Slot size (mm) | Slot velocity (m s⁻¹)* | Approach velocity (m s⁻¹)* | Species | Description |
|---|---|---|---|---|---|
| Canada | **2.54** | 0.06 | **0.04** | "Species not at risk" | Interim code of practice: end-of-pipe fish protection screens for small water intakes in freshwater (FOC, 2021) |
| USA - Virginia | **1** | 0.08 | 0.03 | "Resident aquatic fauna" | Surface water withdrawal intake design and operation standards (SWIWG, 2021)[A] |
| New Zealand | **1.5**[B] | 0.12[C] | **not used**[D] | Native freshwater fish, juveniles and adults, plus introduced salmonids. | Toward National guidance for fish screen facilities (NIWA, 2023) |
| USA - California | **1.75** | 0.12 | **0.06** | Salmonids: "primarily young fish, fish with poor swimming capabilities, and larvae", plus delta smelt. | Anadromous salmonid passage design manual (NMFS, 2022) |
| USA - New York | **0.5-0.75** | **0.15** | 0.04 | Fish (not defined), horseshoe crab, lobster, crayfish, crabs, and shrimp. | Best Technology Available (BTA) for cooling water intake structures (Martens, 2011)[A] |
| USA - California | **1** | **0.15** | 0.06 | All forms of marine life. | California Ocean Plan (SWRCB, 2019)[E] |
| USA - National | **≤2** | **0.15** | 0.08 | All fish and shellfish. Focus on listed species. | Clean Water Act 316(b) - Entrainment (USFWS and NOAA, 2014)[F] |
| USA - National | **9.5** | **0.15** | 0.10 | All fish and shellfish. Focus on listed species. | Clean Water Act 316(b) - Impingement (USFWS and NOAA, 2014)[G] |
| Australia | **2-3** | 0.17 | **0.10** | Native freshwater fish, juveniles and adults. | Design specifications for fish-protection screens in Australia (Boys, 2021) |
| USA - West Coast | **1.75** | 0.20 | **0.10** | Anadromous salmonids. | Fish protection screen guidelines for Washington State (Norlund and Bates, 2000) |
| United Kingdom | **1-2** | 0.28 | **0.10** | *Anguilla anguilla,* all life stages. | (The Eels (England and Wales) Regulations, 2009)[H] |

*Bold values are data from the cited documentation. Non-bold values are derived from those data. Approach velocity was calculated as slot velocity multiplied by the proportion of open area. Slot velocity was calculated as approach velocity divided by the proportion of open area. These conversions assume a 1.75 mm width for wedge wire, which provides the following proportion open area: 0.5 mm slot size (0.22 open area); 0.75 mm (0.30 open area); 1 mm (0.36 open area); 2 mm (0.59 open area); and 9.5 mm (0.84 open area). Where a range of mesh sizes were provided by the cited documentation, the mid-point of value range was used for the conversion. [A]Flexible, subject to permit writer best professional judgement. [B]2 mm if >3 km from coast but varies among jurisdictions. [C]0.06 m/s⁻¹ if no self-cleaning mechanism fitted. [D]Through-screen velocity adopted due to native fish behaviours. [E]Assumes subsurface intake unfeasible. [F]Slot size flexible subject to permit writer best professional judgement. [G]Assumes no entrainment requirements. [H]Screen angle 22-90 degrees; larger slot sizes allowed for larger eels.

Australia provides a timely and relevant case to extend this research globally and therefore advance understanding of larval fish interactions with screens. A major program is now underway in the Murray–Darling Basin to retrofit fish screens to irrigation diversions, aiming to reduce the loss of native fish at unscreened offtakes (Rayner et al., 2024, 2025). While many species and life stages are vulnerable to entrainment at Australian water diversions (Boys et al., 2021b), larval losses in particular can be substantial. Field studies have recorded up to 28 Murray cod larvae per megalitre at unscreened sites (Baumgartner et al., 2007), with millions potentially lost over a single irrigation season (Boys et al., 2021b; Gilligan and Schiller, 2003). Australian guidelines introduced in 2021 recommend a maximum approach velocity of 0.1 m s⁻¹ and a slot width of 2–3 mm (Boys, 2021). These criteria have been shown to reduce entrainment of juvenile and small-bodied fish in both laboratory and field trials (Boys et al., 2013a,b; Bretzel et al., 2025, 2023, 2024; Stocks et al., 2019). However, larvae were not assessed in these studies, and it remains unclear whether the same criteria provide adequate protection for this most vulnerable life stage.

This study addressed that gap by evaluating the effectiveness of modern fish screens in protecting drifting larvae under controlled hydraulic conditions. Murray cod (*Maccullochella peelii*) were used as a model species because their drifting early life stages are representative of many freshwater fish that spawn in regulated rivers with irrigation diversions. After hatching, larvae remain near the nest site for a short period before drifting downstream for approximately 5–7 days (Humphries, 2005). During this phase, they rely on a combination of passive drift and active dispersal, influenced by flow velocity and habitat conditions (Butler et al.,

2011; Lechner et al., 2016), and gradually develop stronger swimming capacity as they age (Kopf et al., 2014). Although Murray cod undergo direct development without a true larval phase (Balon, 1984; Humphries, 2005), the term 'larvae' here refers to individuals from 1 day post hatch (DPH) until they are fully finned and scaled (~30–35 DPH; Neira and Miskiewicz, 1998).

Using 25-day-old larvae, we assessed how approach velocity, wedge wire slot width, and proximity to the screen influenced entrainment rates in a large, hydraulically calibrated flume. By combining a biological assessment with high-resolution three-dimensional flow measurements, we evaluated the efficacy of current screen design criteria and how hydrodynamic conditions – particularly approach velocity, sweeping velocity, and downwelling – affect larval risk. The design parameters tested reflect Australian standards but are comparable to, and in some cases less stringent than, those used in the USA, New Zealand, and the UK. As such, the findings offer broad relevance for screen design in regulated lowland rivers globally, particularly where larval drift coincides with water extraction.

## RESULTS
### Hydraulic conditions in the flume
#### Hydraulics in the presence of a screen
Hydraulic measurements confirmed that prescribed approach velocities of 0.1 and 0.2 m s⁻¹ were reliably achieved at 8 cm in front of the wedge-wire screen face (Fig. 1A, dashed line), validating the accuracy of the experimental setup. At this location – directly relevant to fish–screen interaction – approach velocities only varied slightly across all three screen segments (1, 2 and 3), affirming the effectiveness of the baffle adjustments and flow calibration procedures. Mean approach

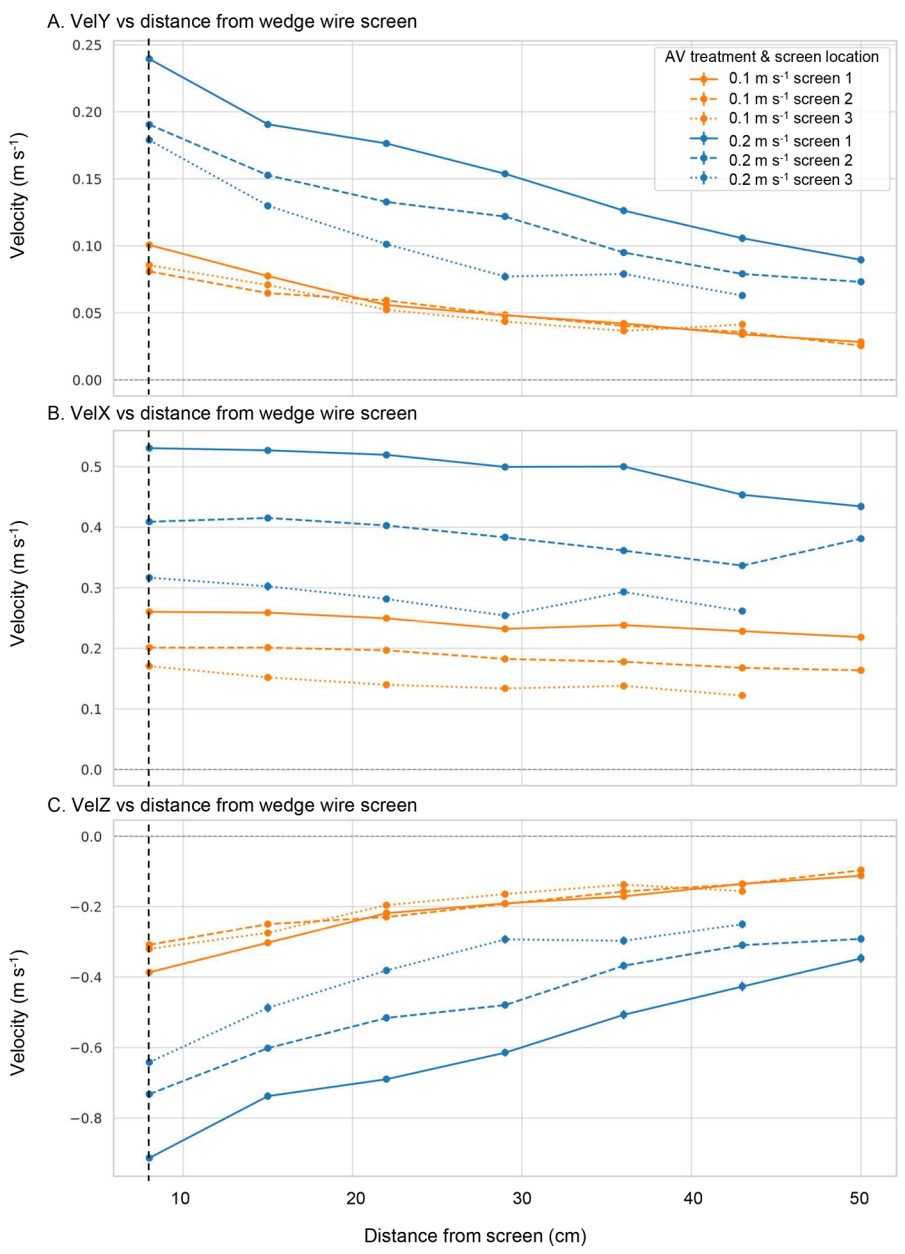

**Fig. 1. Three-dimensional velocity profiles (mean and s.e.m.) in the FlowLab measured at increasing distances perpendicular to a 2 mm wedge-wire screen.** Measurements were taken along three transects positioned at the longitudinal midpoint of each screen segment. Orange lines represent the 0.1 m s$^{-1}$ treatment; blue lines represent the 0.2 m s$^{-1}$ treatment. VelY represents flow directed perpendicular and toward the screen face. VelY at 8 cm from the screen (shown by the dashed vertical line) indicates the achieved approach velocity for each treatment. VelX represents sweeping velocity (parallel to the screen face), and VelZ is the vertical flow component, with negative values indicating downwelling (flow from top to bottom).

velocities across the screen were 0.089 m s$^{-1}$ (±0.006 s.e.m.; range 0.081–0.101) for the 0.1 m s$^{-1}$ treatment, and 0.203 m s$^{-1}$ (±0.019 s.e.m.; range 0.179–0.239) for the 0.2 m s$^{-1}$ treatment, confirming close alignment with target conditions. Corresponding slot velocities, estimated using the equation provided in Eqn S1, were 0.169 m s$^{-1}$ (±0.011 s.e.m.; range 0.011–0.192) for the 0.1 m s$^{-1}$ treatment, and 0.386 m s$^{-1}$ (±0.036 s.e.m.; range 0.340–0.454) for the 0.2 m s$^{-1}$ treatment.

VelY, the component perpendicular to and directed toward the screen face, decreased steadily beyond 8 cm from the screen as the intake's suction influence weakened with distance (Fig. 1A). This decline was steeper in the 0.2 m s$^{-1}$ treatment. VelY was consistently highest at screen 1 (the upstream screen), and lowest at screen 3 (downstream), indicating some minor spatial variation along the intake face. Given that VelY represents the primary flow vector pulling fish toward the screen face, entrainment risk is expected to be greatest for the 0.2 m s$^{-1}$ treatment and at the upstream end of the intake, where approach velocities are strongest.

Sweeping velocity (VelX), representing flow parallel to the screen face, decreased along the intake from screen 1 (upstream) to screen 3 (downstream) (Fig. 1B). This trend reflects the progressive diversion of water through the intake screens, which reduces the volume of flow continuing along the bypass channel. Although the narrowing flume cross-section helps to compensate for this loss by concentrating the remaining flow, it does not fully offset this reduction. Minor backwater effects caused by the downstream fish collection net may also have contributed to the decline in longitudinal velocity. At each screen segment, VelX remained relatively consistent across the perpendicular measurement stations (i.e. at increasing distances from the screen), indicating that sweeping flow was broadly uniform across the channel width but declined longitudinally along the intake. This sweeping velocity defines how fish may be displaced laterally across the screen and shapes the longitudinal transport conditions that can reduce entrainment.

Vertical velocity (VelZ) was predominantly negative across all treatments, indicating downwelling in front of the screen (Fig. 1C).

This vertical flow component was strongest near screen 1 and in the 0.2 m s$^{-1}$ treatment, and it decreased with increasing distance from the screen face – suggesting that downwelling is primarily concentrated close to the screen where suction is greatest. While VelZ is not the primary vector responsible for entrainment, it will undoubtedly compound the risk when combined with VelY, particularly for surface-drifting larvae. As fish are pulled vertically downward, their position in the water column shifts closer to the screen face, potentially increasing contact likelihood and reducing opportunities to escape entrainment pathways.

### Hydraulics in the absence of a screen

In the absence of the wedge-wire screen, approach velocities (VelY) increased substantially across all screen segments and flow treatments, with values at 8 cm in front of the intake notably higher than in the screened condition (Fig. 2A). Sweeping velocity (VelX) was similar

between the unscreened and screened treatments under the 0.1 m s$^{-1}$ condition but substantially elevated under the 0.2 m s$^{-1}$ treatment. As with the screened condition, VelX remained relatively consistent across the channel width and decreased progressively downstream (Fig. 2B).

The spatial extent of the suction field also changed. Under screened conditions, the prescribed AVs of 0.1 and 0.2 m s$^{-1}$ were reached approximately 8 cm from the screen face (Fig. 1A, dashed line). In contrast, under unscreened conditions, equivalent velocities were not reached until approximately 15 cm (0.1 m s$^{-1}$ treatment) and 30 cm (0.2 m s$^{-1}$ treatment) from the intake (Fig. 2A). This indicates a broader zone of influence when the screen is absent.

Vertical velocities (VelZ) were strongly negative (i.e. downwelling) near the intake (Fig. 2C), especially at screen segment 3 and under the 0.2 m s$^{-1}$ treatment. The intensity of downwelling decreased with distance from the intake and was more pronounced than in the screened configuration.

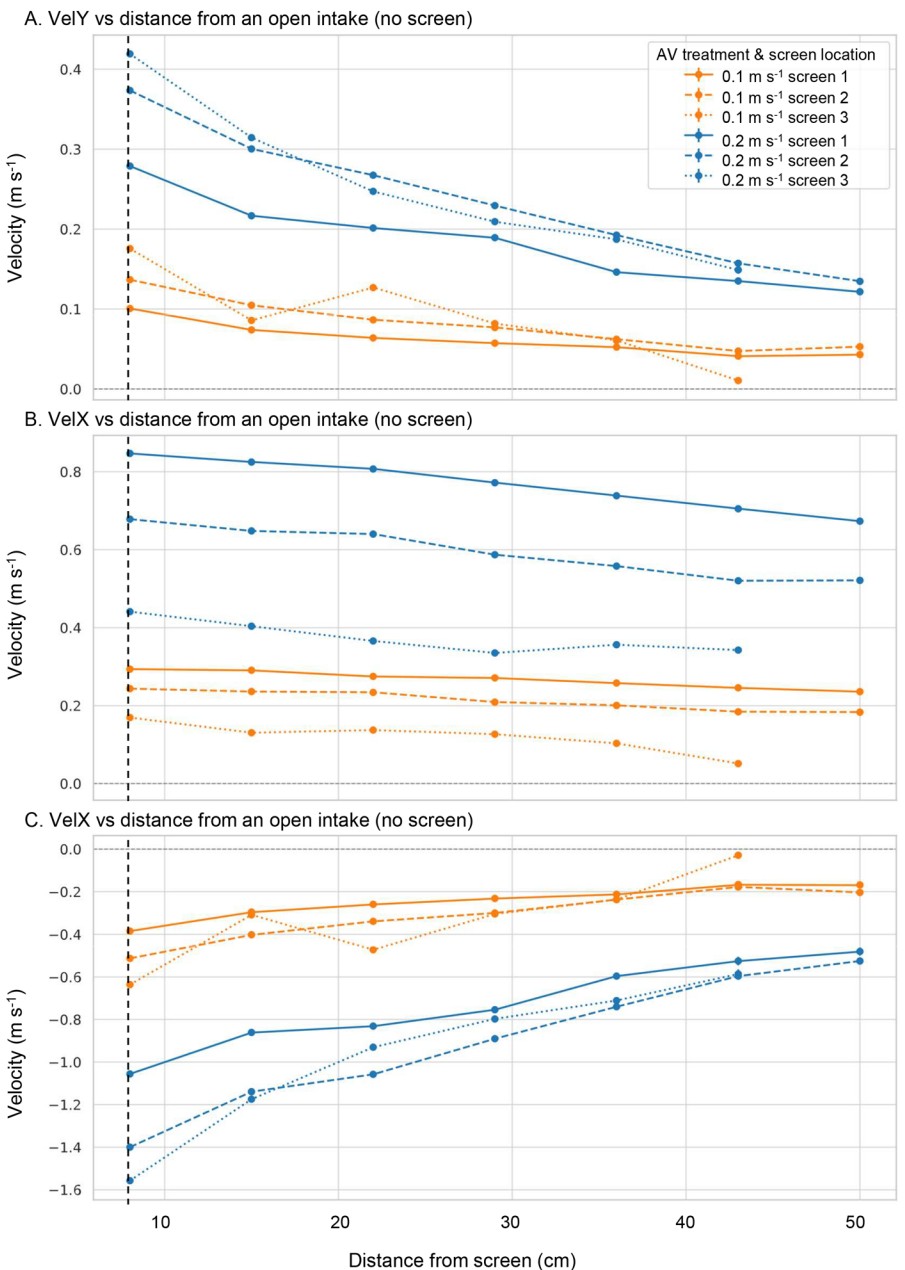

**Fig. 2. Three-dimensional velocity profiles (mean and s.e.m. *n*=80) in the FlowLab measured at increasing distances perpendicular to the intake when the screen was not present.** Measurements were taken along three transects positioned at the longitudinal midpoint of each intake segment. Orange lines represent the 0.1 m s$^{-1}$ treatment; blue lines represent the 0.2 m s$^{-1}$ treatment. VelY represents flow directed perpendicular and toward the intake. VelY at 8 cm from the intake (shown by the dashed vertical line) indicates the achieved approach velocity for each treatment. VelX represents sweeping velocity (parallel to the intake), and VelZ is the vertical flow component, with negative values indicating downwelling (flow from top to bottom).

### Larval entrainment patterns across treatments

Larval entrainment was highest in the unscreened treatments, with mean probabilities of 90%, 78% and 60% for the near, mid and far release distances, respectively (Fig. 3). Adding a wedge-wire screen significantly reduced entrainment when compared to the unscreened intake: by as much as 94% when a 2 mm slot and a 0.1 m s$^{-1}$ approach velocity were combined (Fig. 3). All three factors (approach velocity, slot size, and release distance) interacted significantly to influence entrainment rates (approach velocity×slot size: $\chi^2$=17.3, d.f.=1, $P$<0.0001; approach velocity×release point: $\chi^2$=10.0, d.f.=2, $P$<0.01; release point×slot size: $\chi^2$ 11.1, d.f.=2, $P$<0.005).

Approach velocity had the largest single effect. Larvae were significantly less likely to be entrained at 0.1 m s$^{-1}$ than at 0.2 m s$^{-1}$ across all slot sizes and release distances (Fig. 3). This effect was strongest when paired with the 2 mm slot size: larvae were 63 times less likely to be entrained at 0.1 m s$^{-1}$ compared to 0.2 m s$^{-1}$ when averaged across release points. At the near release distance, this difference increased to a 200-fold reduction in entrainment odds (Table 2). This suggests that the combination of lower approach velocity and finer slot width can markedly reduce the vulnerability of drifting larvae, especially when they are close to the screen.

Reducing slot size from 3 mm to 2 mm also reduced entrainment, though the effect was smaller than that of approach velocity. At 0.1 m s$^{-1}$, the finer slot reduced entrainment by 11 times compared to the 3 mm slot. However, this benefit diminished at the higher approach velocity (0.2 m s$^{-1}$), where larvae were equally likely to be entrained regardless of slot size (odds ratio ≈1) (Table 2). This interaction indicates that reducing slot size may be less beneficial at higher approach velocities.

Proximity to the intake strongly influenced entrainment in unscreened treatments, with larvae released near the intake being entrained at significantly higher rates than those released mid or far. With screens in place, this pattern was moderated but still present across most treatments. In general, larvae released at the mid and far points were 1.9–2.2 times less likely to be entrained than those released near the screen ($P$<0.05). However, when the 2 mm slot was combined with 0.1 m s$^{-1}$, proximity had little effect and entrainment remained uniformly low (Fig. 3 and Table 2). This indicates that with this optimal combination of screen parameters, larval fish can approach very close to the screen surface and still have a high probability of avoiding entrainment.

## DISCUSSION
### Effect of approach velocity and slot size on larval entrainment

This study found that both approach velocity and slot size had significant and interacting effects on larval entrainment. Larvae were far less likely to be entrained at an approach velocity of 0.1 m s$^{-1}$ than at 0.2 m s$^{-1}$, regardless of slot size. Slot size also influenced outcomes, though the magnitude of its effect was lower and context dependent. At 0.1 m s$^{-1}$, reducing slot size from 3 mm to 2 mm resulted in an 11-fold reduction in entrainment, but this benefit was not observed at 0.2 m s$^{-1}$. These results highlight the primacy of approach velocity in reducing entrainment and suggest that slot size is most effective under lower velocity conditions. When combined, the most protective condition (2 mm slot and 0.1 m s$^{-1}$ velocity) reduced entrainment by up to 94% compared to the unscreened treatment, with rates dropping to as low as 0–5%.

Importantly, even when larvae were released in close proximity to the screen under this optimal configuration, entrainment remained low, indicating that the effective zone of entrainment was tightly confined to the immediate screen face. This suggests that the combination of low approach velocity and fine slot size substantially limits the screen's zone of influence, allowing larvae to approach close to the structure without being drawn in.

A valuable point of comparison is provided by a previous laboratory evaluation of wedge-wire cylindrical screens (EPRI, 2003), which assessed entrainment and impingement outcomes across early life stages of eight North American fish species: Striped bass (*Morone saxatilis*), Winter flounder (*Pseudopleuronectes americanus*), Yellow perch (*Perca flavescens*), Rainbow smelt (*Osmerus mordax*), Common carp (*Cyprinus carpio*), White sucker (*Catostomus commersonii*), Alewife (*Alosa pseudoharengus*), and Bluegill (*Lepomis macrochirus*). Despite some study difference in screen type (cylindrical versus flat-panel) and inclusion of impingement as a response variable, the slot velocities in that study (0.15 and 0.30 m s$^{-1}$) closely match the slot velocities derived in our study (0.17 and 0.39 m s$^{-1}$), enabling meaningful cross-study comparison. Considerable variation in impingement and entrainment rates was reported among species and test conditions. For instance, White sucker and Striped bass larvae experienced entrainment rates of 10–60%, Common carp larvae 20–70%, yellow perch 10–50%, and Bluegill and Rainbow smelt over 80% in some trials.

Our results reveal that Murray cod larvae show a similarly wide range of susceptibility, with entrainment as high as 80% under the least protective configuration (3 mm slot, 0.2 m s$^{-1}$ approach velocity). However, under the most protective tested screen design (2 mm slot, 0.1 m s$^{-1}$ approach velocity), entrainment of Murray cod dropped to 0–5%, representing a 94% reduction compared to unscreened conditions. These results suggest that, when optimally

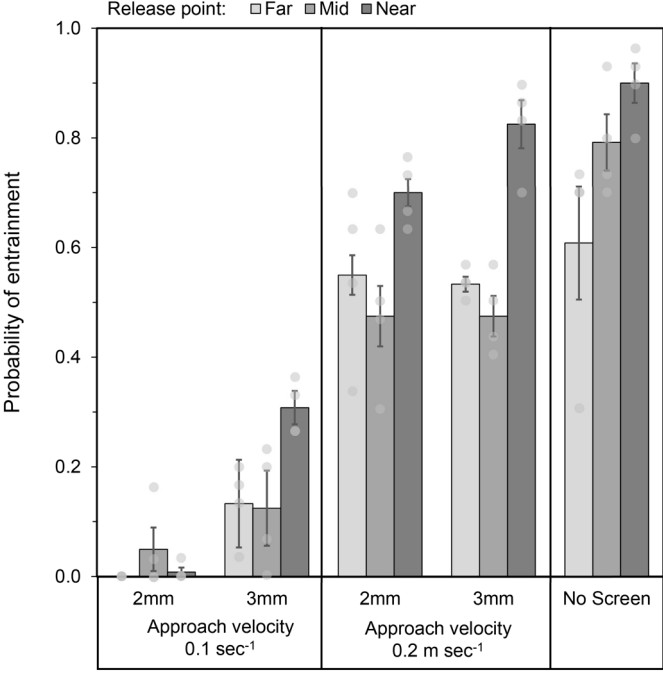

**Fig. 3. Modelled probability of entrainment (±95% Wald confidence intervals) for Murray cod larvae at flat panel wedge wire screens in the experimental flume.** The plots show the interaction between slot size (2 mm and 3 mm), approach velocity (0.1 and 0.2 m s$^{-1}$) and release points relative to the screen (grey shading, near, mid and far). Circle markers show spread of values (*n*=4) used to calculate the means.

**Table 2. Complete list of odds ratios for all comparisons and treatment levels**

| Comparison | Interacting factor | | Odds ratio[a] | 95% Wald confidence limits | | Interpretation of interaction between the factors[b] |
|---|---|---|---|---|---|---|
| Approach velocity: 0.1 m s⁻¹ versus 0.2 m s⁻¹ | 2 mm slot | Far | **0.003** | <0.001 | 0.057 | The 0.1 m s⁻¹ approach velocity is always less likely to entrain than 0.2 m s⁻¹. |
| | | Mid | **0.063** | 0.026 | 0.15 | The 2 mm mesh size is always less likely to entrain than 3 mm. |
| | | Near | **0.005** | 0.001 | 0.029 | The release point effect is inconsistent across the entire channel, but the highest |
| | 3 mm slot | Far | **0.138** | 0.073 | 0.26 | entrainment rates for each velocity and slot size were generally at the near |
| | | Mid | **0.162** | 0.085 | 0.309 | release point. |
| | | Near | **0.097** | 0.053 | 0.178 | |
| Slot size: 2 mm versus 3mm | 0.1 m s⁻¹ | Far | **0.026** | 0.002 | 0.449 | 0.1 m s⁻¹ approach velocity is always less likely to entrain. |
| | | Mid | 0.386 | 0.148 | 1.006 | 2 mm mesh size less likely to entrain at near release point. |
| | | Near | **0.028** | 0.005 | 0.147 | Release point effect is inconsistent, but near generally entrains more. |
| | 0.2 m s⁻¹ | Far | 1.069 | 0.643 | 1.776 | |
| | | Mid | 1 | 0.602 | 1.66 | |
| | | Near | **0.5** | 0.272 | 0.92 | |
| Release points: Far versus mid | 0.1 m s⁻¹ | 2 mm | 0.073 | 0.004 | 1.328 | 0.1 m s⁻¹ approach velocity nearly always less likely to entrain. |
| Far versus near | | | 0.331 | 0.013 | 8.306 | 2 mm mesh size usually less likely to entrain. |
| Mid versus near | | | 4.523 | 0.747 | 27.39 | Far release point always less likely to entrain than near. But mid release point |
| Far versus mid | | 3 mm | 1.075 | 0.509 | 2.269 | inconsistent entrainment rates |
| Far versus near | | | **0.352** | 0.184 | 0.673 | |
| Mid versus near | | | **0.327** | 0.169 | 0.633 | |
| Far versus mid | 0.2 m s⁻¹ | 2 mm | **1.348** | 0.811 | 2.239 | |
| Far versus near | | | **0.527** | 0.31 | 0.896 | |
| Mid versus near | | | **0.391** | 0.23 | 0.664 | |
| Far versus mid | | 3 mm | 1.261 | 0.759 | 2.093 | |
| Far versus near | | | **0.247** | 0.137 | 0.445 | |
| Mid versus near | | | **0.196** | 0.109 | 0.353 | |

[a]Bolded indicates statistically significant (*P*>0.05).
[b]Interpretation discusses the relationships of the individual odds ratio values between the treatment combinations. For example, considering the shaded odds ratios below. All ratios are <1 indicating that overall, 0.1 m s⁻¹ is less likely to entrain than 0.3 m s⁻¹ for any combination of slot size or release point. Next, consider that in the same shaded section, all of the 2 mm odds ratios are lower than all of the 3 mm odds ratios, indicating that the influence of velocity is different for each slot size. That is, the 2 mm slot size results in less entrainment than the 3 mm slot size.

configured, screens can afford substantially higher levels of protection than previously reported for many other larval species.

Our conclusions also echo earlier syntheses of entrainment impacts. Chow et al. (1981) reviewed numerous studies from the 1970s and early 1980s and similarly identified approach velocity as the primary determinant of larval outcomes, and most notably that thresholds above ~0.15 m s⁻¹ substantially increased susceptibility. The review noted that while early evaluations of fine-mesh and wedge-wire screens provided some reduction in larval entrainment, effectiveness was often limited at higher velocities or when larvae lacked strong swimming capacity. The consistency between those early findings and the present study underlines that velocity remains the dominant factor shaping entrainment risk, while finer-scale design criteria such as slot size become most valuable under lower velocity conditions.

A final note concerns slot size. Earlier studies assumed that screens must have mesh openings smaller than the body dimensions of fish to be protective (Schneeberger and Jude, 1981). Our results, consistent with other laboratory evaluations (EPRI, 2003; Heuer and Tomljanovich, 1978), show that this is not necessarily the case. The Murray cod larvae tested here (mean body depth 2.9 mm; width 2.1 mm) were physically small enough to pass through the 3 mm slot width. Yet entrainment rates at this slot size remained relatively low (~10–30%) when approach velocity was limited to 0.1 m s⁻¹. This finding underscores that approach velocity is the dominant factor, and that mesh dimensions cannot be considered in isolation from the hydraulic conditions at the screen face. Put simply, if flows are weak enough for larvae to detect and resist, even small and weak-swimming individuals can avoid entrainment at openings larger than their body size.

## Hydrodynamic mechanisms underpinning entrainment patterns

The hydraulic structure created by the wedge-wire screen generated a dynamic and spatially variable flow field that would be expected to influence larval entrainment patterns in several ways. Approach velocities (VelY), which define the vector pulling larvae directly toward the screen face, were greatest at the upstream end (screen 1) and in the higher flow treatment (0.2 m s⁻¹), where local suction was strongest. As VelY declined rapidly with distance from the screen, larvae located further into the channel had lower probability of being drawn in, suggesting a strong spatial gradient in entrainment risk.

Sweeping velocity (VelX), representing flow parallel to the screen face, decreased along the intake from screen 1 (upstream) to screen 3 (downstream), reflecting the progressive removal of water through the intake structure (Fig. 1B). Although the narrowing flume cross-section helped concentrate the residual bypass flow, this was not sufficient to fully offset the volume loss, resulting in a clear longitudinal reduction in sweeping velocity. At each screen segment, VelX remained relatively consistent across the channel width, indicating uniformity in the lateral direction.

While sweeping flow is often viewed as a mechanism that helps move fish past the screen more rapidly, potentially reducing contact time and exposure (EPRI, 2003; Swanson et al., 2004), this can be diminished for smaller fish with weaker swimming ability when they are close to the screen and exposed to elevated approach velocity (EPRI, 2003). It is likely that we observed this in the current study. In the 0.2 m s⁻¹ treatment, the increased magnitude of VelX likely overwhelmed the swimming capacity of surface-drifting larvae, reducing their ability to maintain position or avoid the intake zone when combined with perpendicular (VelY) and vertical (VelZ)

vectors drawing larvae toward the screen face. Therefore, although VelX provides transport along the screen face, its elevated magnitude under the higher flow conditions tested may have done little to provide additional protection and may actually have been a contributing vector for entrainment rather than a mitigating one.

Vertical velocity (VelZ) was characterised by downwelling in front of the screen, strongest near screen 1 and in the 0.2 m s$^{-1}$ treatment, and diminishing with increasing distance from the screen face. Although VelZ plays a lesser role than VelY in directly driving larvae into the screen, its downward pull shifts drifting larvae from the surface deeper into the flow field, bringing them into closer proximity with the screen face and regions of higher approach velocity, particularly under stronger drawdown.

Together, the three velocity components define a spatially variable environment, not unlike that experienced in natural river scenarios. Larvae drifting near the upstream end of the intake are likely to encounter the strongest perpendicular pull (VelY), the fastest sweeping flow (VelX), and the most pronounced downwelling (VelZ), all of which may act together to increase entrainment likelihood. These risks are further amplified under the 0.2 m s$^{-1}$ treatment, where all velocity components were stronger and more spatially variable, reinforcing the expectation of higher entrainment potential. In contrast, under the 0.1 m s$^{-1}$ treatment, reduced approach velocities, combined with weaker downwelling and less energetic sweeping flow, may offer greater opportunity for larvae to avoid or resist entrainment. These hydraulic differences provide a clear mechanistic explanation for treatment effects observed in larval entrainment outcomes.

In contrast, when the wedge-wire screen was removed, the hydraulic environment surrounding the intake changed substantially, with direct implications for larval entrainment. Approach velocities (VelY) were substantially higher across all screen segments, particularly under the 0.2 m s$^{-1}$ treatment, which would intensify the force pulling larvae toward the intake. Sweeping velocities (VelX) were also elevated in this higher flow condition, suggesting an increased potential for weak swimming larvae to be transported uncontrollably along the intake face, becoming entrained in perpendicular and vertical flow vectors. Notably, the zone over which the intake exerted a hydrodynamic influence expanded in the absence of a screen: target approach velocities were reached much further from the intake (15–30 cm depending on treatment), effectively enlarging the suction field and increasing the spatial window within which larvae could be entrained. This broader zone of influence, combined with stronger downwelling (VelZ) near the intake, particularly at screen segment 3, indicates that larvae released near the surface would be more rapidly drawn downward and toward the diversion channel in unscreened conditions. Collectively, these shifts in the velocity field suggest that screens not only reduce absolute approach velocities but also contract the zone of influence, both of which are likely to reduce the probability and spatial scale of larval entrainment.

### Conclusion and implications for design standards

This study provides new empirical evidence that Australia's modern fish screen design criteria, originally developed with juvenile fish in mind, can also offer substantial protection to larval stages. Specifically, the tested combination of a 0.1 m s$^{-1}$ approach velocity and 2–3 mm slot widths reduced Murray cod larval entrainment by up to 94% compared to unscreened conditions. These results align with international findings and demonstrate that well-designed, hydraulically calibrated screens can deliver protection across multiple life stages, even for larvae with smaller body size and weaker swimming ability.

However, the findings also highlight how sensitive entrainment can be to even modest changes in design. Increasing approach velocity from 0.1 m s$^{-1}$ to 0.2 m s$^{-1}$, or widening slot size from 2 mm to 3 mm, resulted in sharp increases in entrainment risk. This reinforces the importance of adhering strictly to prescribed specifications if they have been verified with biological testing. Where practical or operational constraints require deviation, the increased risk to early life stages must be carefully weighed.

Internationally, some jurisdictions adopt more conservative standards, such as slot sizes down to 1 mm and approach velocities as low as 0.03 m s$^{-1}$ (Table 1), sometimes driven by the presence of highly vulnerable species like glass eels or elvers (Carter et al., 2023; Jellyman et al., 2023). While such conservative benchmarks may further reduce entrainment, meeting them they may have significant design and cost implications. This underscores the value of targeted empirical assessments to inform screen design, allowing protection guidelines to be tailored to local species and flow conditions while balancing engineering feasibility. Where early life stage interactions remain poorly understood, further research is warranted to refine and validate design criteria under real-world operating conditions.

## MATERIALS AND METHODS

### Fish collection and care

Murray cod eggs were harvested from spawning boxes in earthen ponds at the Narrandera Fisheries Centre, New South Wales, Australia. Eggs were incubated for 7–10 days until hatching. Larvae were maintained in flow-through trays supplied with river water and aerated with air stones. From 17 DPH, larvae began exogenous feeding on newly hatched *Artemia* nauplii and pulverised commercial pellets.

At 23 DPH, larvae were transported in oxygenated bags within insulated boxes to maintain stable temperatures during the ~3-h journey to the test facility. Upon arrival, larvae were acclimated in flow-through trays (~10 l min$^{-1}$) connected to the flume reservoir. Trays were aerated, and larvae were fed daily. Healthy fish were selected for experiments at 25 and 26 DPH. Approximately 2–3 h before entering the flume, 30 larvae were pipetted into a 700 ml jar aerated via vinyl tubing, with multiple jars corresponding to different replicated treatment groups.

To obtain daily morphometric measurements of the fish being tested, a random subsample of 20 larvae was selected from the main population and euthanised. For the duration of the experiment these individuals measured (mean±s.e.): total length 13.2±0.3 mm, dorso–ventral depth 2.9±0.1 mm, and maximum width 2.1±0.0 mm. Water quality in holding trays was monitored twice daily and remained within optimal ranges: pH 7.8±0.2 (7.2–8.1), conductivity 0.2±0.0 mS cm$^{-1}$, dissolved oxygen 9.1±1.1 mg l$^{-1}$ (7.3–12.4), total dissolved gas saturation 101.5±6.7% (92–122), and temperature 18.6±1.2°C (15.2–21.0).

The use of animals in this research and experimental protocols were approved by the Charles Sturt University and NSW Department of Primary Industries Fisheries Animal Ethics Committees under CSU protocol A19253 and NSW DPI ACEC REF 19/01. A Victorian Fisheries stocking and translocation permit exemption was obtained to allow hatchery fish from NSW to be used in the Cohuna FlowLab in Victoria (2019-19NC).

### Flume design and operation

Experiments were conducted at the FlowLab (AWMA Water Control Solutions, Cohuna, Victoria), a high-capacity flume designed to simulate realistic open-channel flow conditions for testing hydraulic infrastructure (Fig. 4A). The facility supports flow rates from 0.5 to 100 ml day$^{-1}$, achieved using a variable-speed diesel pump circulating water from a reservoir. Electromagnetic flow meters (Aquamonix, Milperra, NSW) monitored intake and discharge flows to ensure calibration. For this study, flow rates up to 42 ml day$^{-1}$ were required.

Water was channelled through a flow straightener into a straight channel, then flowing into a 12 m long×2.4 m wide×1 m deep screen-testing flume (Fig. 4A). The flume was divided into two parallel channels: one where

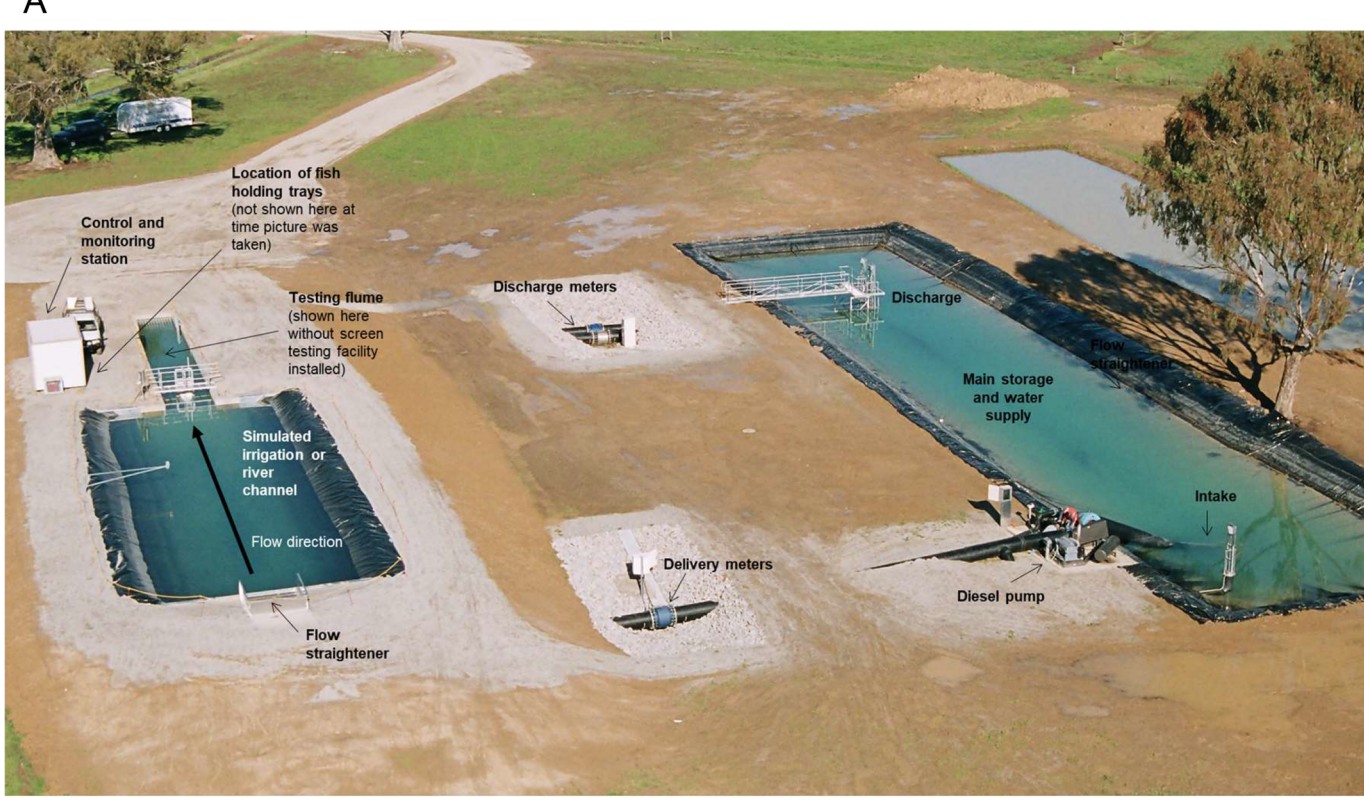

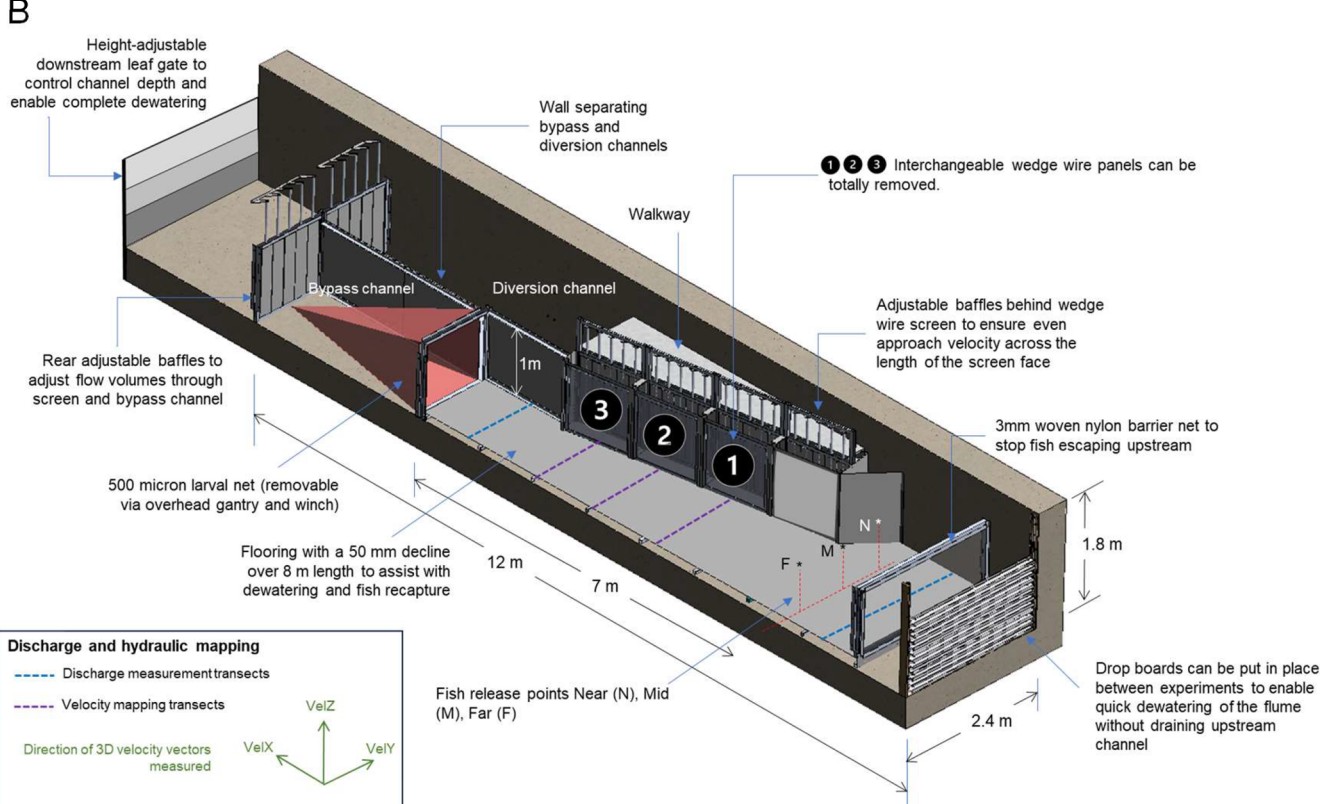

**Fig. 4. Experimental screen testing facility.** (A) Aerial photo of the FlowLab calibrated volumetric test facility. (B) Isometric projection of the testing flume. Shown are its major components and dimensions, as well as the transects and vectors used to take hydraulic measurements using the three-dimensional acoustic Doppler velocimeter. The test screens (1–3) could be operated with 2 mm or 3 mm wedge-wire panels in place, or totally removed to leave an open intake for the no screen treatment.

water flowed adjacent to and past the screens (bypass channel), and the other where water flowed through a screened or unscreened diversion (diversion channel). Adjustable gates and baffles at the downstream end of each channel allowed precise control of flow 'shunting' from the bypass into the diversion channel while maintaining flume depth at 0.6 m regardless of discharge (Fig. 4B).

The screening system comprised three interchangeable 1 m² stainless steel wedge-wire panels (totalling 3 m² total screen face). Wedge-wire width was 1.8 mm and the screens tested were either 2 mm slot width (52.6% open area) or 3 mm slot width (62.5% open area). Approach velocity (the water speed approaching perpendicular to the screen face, measured 8 cm upstream; Boys et al., 2021a) was set to either 0.1 or 0.2 m s$^{-1}$ by adjusting pump speed. Baffles behind the screens were adjusted to control flow through each screen and hence helped achieve a uniform approach velocity across the length of the screen. A 500 µm net installed at the bypass outlet collected larvae that were not entrained (Fig. 4C). A 3 mm exclusion screen at the upstream end prevented fish escape.

### Hydraulic calibration and mapping
#### Instrumentation
All hydraulic measurements were conducted using a three-dimensional acoustic Doppler velocimeter (Sontek FlowTracker2 Lab ADV®, Xylem Inc., New York) and associated FlowTracker2 software (version 1.6), operated in accordance with the manufacturer's guidelines (Xylem, 2019). The instrument resolves three orthogonal velocity components (VelX, VelY, and VelZ) corresponding to the longitudinal, lateral, and vertical axes relative to the probe orientation. In this study, the probe was aligned so that VelY, represented the component perpendicular to and directed toward the screen face. At 8 cm in front of the screen face VelY corresponded to the approach velocity being tested. VelX captured the velocity running parallel to the screen face in the upstream–downstream direction (often referred to as the sweeping velocity). Finally, VelZ represented the vertical flow, with negative values indicating downward movement (Fig. 4B). The probe was mounted on a custom XYZ positioning system fixed above the flume, enabling precise and repeatable placement throughout the channel.

The FlowTracker2 operates by emitting high-frequency acoustic pulses (ping rate: 40 Hz) and measuring Doppler shifts from suspended particles in the water column. Each velocity sample is the average of 20 pings, providing a stable estimate of instantaneous 3D velocity and associated signal metrics. The instrument records two samples per second (2 Hz). At each measurement point, 80 samples were recorded over a 40-second interval and used to calculate the mean and standard error of velocity. Each sample also includes a signal-to-noise ratio (SNR), used to assess signal strength and interference. Both SNR and standard error were monitored in real time, and only measurements that met SonTek's recommended quality control thresholds were retained.

#### Confirming correct approach velocity
Prior to experimental trials, the flume was hydraulically calibrated to achieve target approach velocities of 0.1 and 0.2 m s$^{-1}$, measured 8 cm in front of the screen face. Calibration was achieved by adjusting inflow rates and fine-tuning baffle positions located downstream of the intake and bypass channels and behind each screen. Measurements were repeated under steady-state conditions until velocity targets were achieved. At each flow setting, velocities were recorded at two depths (0.2 and 0.8 of local depth) at the longitudinal midpoint of each screen segment (screens 1, 2 and 3; see Fig. 4B). These depth-specific values were averaged to yield a representative approach velocity for each screen. Calibration continued iteratively until consistent and spatially uniform AV values were achieved and could be reliably maintained across treatments.

Because many international studies and guidelines report slot velocity rather than approach velocity, we derived slot velocity values from our known approach velocities using Eqn S1. This enables broader comparison of our results with existing global literature, where velocity thresholds and fish performance criteria are often based on flow through the screen slot rather than in front of it.

#### Mapping hydraulic conditions throughout the flume
To characterise the flow field further away from the screen face, point-based 3D velocity measurements were taken across three transects extending

perpendicularly from screens 1, 2 and 3 (Fig. 4B). Measurements were collected at up to seven locations across those transects, beginning 8 cm from the screen face and extending outward at 50 cm intervals. In treatments where the screen was removed, measurements were taken at the same locations, with the first taken 8 cm in front of the plane where the screen would have been. At each location, velocity was recorded at two depths (0.2 and 0.8 of the local depth), and the average was used to represent that point. This design enabled spatial profiling of the hydraulic field both longitudinally downstream along the flume and laterally across the intake. These velocity fields were plotted in each of the X, Y and Z vectors to describe how the velocity profile of the flume changed in the longitudinal and lateral planes.

### Determining bypass and diversion channel discharge
Discharge measurements were taken for two primary purposes. First, to rule out whether differences in larval entrainment rates between treatments could be attributed to variations in the relative flow partitioning between the intake and bypass channels. Intake discharge was used to correct for discharge and it was confirmed that expressing entrainment as a function of discharge did not alter the relative ranking of entrainment risk across treatments. Therefore, the decision was made to only present absolute entrainment rates throughout this paper to support clear interpretation and avoid duplication and unnecessary confusion. Second, reporting the flow conditions provides a benchmark for replicating the trials in future experiments involving different species or life stages.

To quantify discharge partitioning, flow was measured at two locations: (1) the upstream inlet of the flume and (2) the downstream end of the bypass channel (Fig. 4B). These measurements provided values for total flume inflow and bypass outflow, with the difference used to calculate the volume of water diverted through the intake. Discharge at each location was calculated using the mean-section method (Xylem, 2019), dividing the channel cross-section into six verticals spaced ∼70 cm apart. At each vertical, velocities were recorded at 0.2, 0.6 and 0.8 of the local water depth and averaged to estimate depth-averaged velocity. Sub-sectional discharge was calculated as the product of velocity, depth, and segment width, and total discharge was determined by summing across all verticals. All measurements were repeated under steady flow conditions.

Because pump rate, gate position, and baffle configuration were identical across mesh treatments, discharge was only measured under the 2 mm wedge-wire screen configuration. These values were considered representative for both 2 mm and 3 mm mesh sizes. Summary discharge values for each treatment (including total inflow, bypass outflow, and calculated intake discharge) are provided in Table 3. Values are expressed in both cumecs (m³ s$^{-1}$) and megalitres per day (ml day$^{-1}$) for reference.

### Experimental design and protocol
Twelve screening treatments were conducted (Table 4), comprising all combinations of two approach velocities (0.1 and 0.2 m s$^{-1}$), two slot sizes (2 and 3 mm), and three release distances (near=0.5 m, mid=1.0 m and

**Table 3. Summary of measured discharge (Q) entering the testing flume, exiting via the downstream end of the bypass channel (bypass discharge), and diverted through the intake under each treatment scenario**

| Treatment | Q entering flume | Q downstream of bypass | Q diversion channel[a] |
|---|---|---|---|
| No screen 0.1 m s$^{-1}$ | 0.25 (21.24) | 0.03 (2.25) | 0.22 (18.99) |
| No screen 0.2 m s$^{-1}$ | 0.54 (46.85) | 0.06 (4.92) | 0.49 (41.93) |
| 2 mm screen 0.1 m s$^{-1}$ | 0.24 (20.38) | 0.05 (4.03) | 0.19 (16.35) |
| 2 mm screen 0.2 m s$^{-1}$ | 0.39 (33.37) | 0.09 (7.69) | 0.30 (25.68) |

Values are presented as cubic cumecs (m³ s$^{-1}$), with corresponding volumes in megalitres per day (ml/day) in parentheses. These measurements were used to validate flow partitioning, confirm consistency between treatments, and provide a reference point for potential replication using other species or life stages.

[a]Diversion channel flow was calculated as the difference between flume inflow and bypass outflow.

**Table 4. Experimental design**

| Slot width | Approach velocity | Release Point | Number of replicates | Fish per replicate |
|---|---|---|---|---|
| No screen | na* | Near | 4 | 30 |
| | na* | Mid | 4 | 30 |
| | na* | Far | 4 | 30 |
| 2 mm | 0.1 ms$^{-1}$ | Near | 4 | 30 |
| | | Mid | 4 | 30 |
| | | Far | 4 | 30 |
| | 0.2 ms$^{-1}$ | Near | 4 | 30 |
| | | Mid | 4 | 30 |
| | | Far | 4 | 30 |
| 3 mm | 0.1 ms$^{-1}$ | Near | 4 | 30 |
| | | Mid | 4 | 30 |
| | | Far | 4 | 30 |
| | 0.2 ms$^{-1}$ | Near | 4 | 30 |
| | | Mid | 4 | 30 |
| | | Far | 4 | 30 |

na* because no screen was present and approach velocity is technically defined 8 cm in front of a screen. The no screen treatment was run at a single diversion velocity which was created by running the flume under the discharge conditions equivalent to the 0.2 m s$^{-1}$ AV treatment.

far=1.5 m from the screen). In addition, three unscreened controls were included, in which no mesh screen separated the bypass and diversion channels. Each treatment was replicated four times, with 30 larvae per replicate.

For each trial, the designated screen and flow conditions were established. Larvae were released upstream at the assigned distance from the screen. After 5 min, flow was stopped, the flume was drained, and the net was retrieved using an overhead winch. Larvae were counted to determine entrainment.

This experimental configuration, including the large scale of the flume and varied release distances, was intended to enable larvae to exhibit natural movement and orientation relative to flow, providing a more realistic test of screen interaction and natural avoidance behaviour.

## Statistical analyses

In this study, entrainment specifically describes the passage of larval fish through the screen mesh, or through the unscreened intake, into the diversion channel, rather than being safely deflected along the bypass flow. Entrainment probability ($P$) was calculated as:

$$P = 1 - \frac{recaptured}{released}$$

In this equation, *released* is the number of larvae introduced into the flume (always 30), and *recaptured* is the number collected in the bypass channel catch net after draining. Values of $P$ range from 0 to 1, where 0 indicates no entrainment (all larvae were recaptured) and 1 indicates complete entrainment (no larvae were recaptured).

During initial tests, it was assumed that some larvae could be lost during draining or handling, independent of screen performance (for example, escaping through small gaps between the net and flume components). To quantify this sampling error, four control trials were conducted in which solid metal sheets replaced the screens, preventing any entrainment. These controls showed that, on average, one larva per trial was not recovered due to procedural losses. This value was treated as the expected background loss unrelated to screening.

Since the number of larvae released was always 30, this constant value was substituted into the adjusted formula. Adding 1 to the number recaptured accounts for the average sampling error, ensuring that the estimated entrainment probabilities more accurately reflect true screen performance rather than handling loss. To incorporate the constant and sampling error, the entrainment probability was calculated using the following equation:

$$P = 1 - \frac{recaptured + 1}{30}$$

A logistic regression model was used to assess the effects of slot size, approach velocity, and release distance, including all interactions. Predicted probabilities and odds ratios (OR) were derived to quantify the likelihood of entrainment under different conditions. OR values below 1 indicated a reduced likelihood of entrainment (e.g. OR=0.33 means the event was three times less likely), while OR values above 1 indicated increased likelihood (e.g. OR=2.4 means 2.4 times more likely).

Unscreened controls were excluded from the regression analysis but were plotted separately for comparison. All analyses were conducted using SAS/STAT 14.1 (SAS Institute Inc., Cary, NC, USA).

## Copy editing

This manuscript was copy edited with the assistance of an AI language model (ChatGPT 4o, OpenAI, 2024) to improve grammar, spelling, consistency and clarity. The authors iteratively reviewed and revised the content to ensure accuracy and accept full responsibility for the final version.

### Acknowledgements

The authors would like to acknowledge the Wiradjuri and Barapa people, the Traditional Owners of the lands where the hatchery fish were bred and where this research was conducted, extending respect to Elders past, present, and future. Special thanks are extended to Brett Kelly and the team at AWMA Water Control Solutions for the use of their FlowLab, as well as for their assistance in designing and constructing the screen-testing facility. The invaluable contributions of technician Tony Fowler and Matthew Balzer, were crucial to the completion of the experiments. Lachie Jess from the Narrandera Fisheries Centre native fish hatchery provided the fish used in this experiment. Table 1 was produced with input by John Burnett, Intake Screens Inc.

### Competing interests

The authors declare no competing or financial interests.

### Author contributions

Conceptualization: C.A.B., W.R., K.E.D., L.J.B.; Data curation: C.A.B., W.R.; Formal analysis: C.A.B., W.R.; Funding acquisition: C.A.B., L.J.B.; Investigation: C.A.B., K.E.D., T.R., P.M.; Methodology: C.A.B., W.R., K.E.D., L.J.B.; Project administration: C.A.B.; Resources: C.A.B., L.J.B.; Supervision: C.A.B., K.E.D., L.J.B.; Validation: C.A.B.; Visualization: C.A.B., W.R.; Writing – original draft: C.A.B., W.R., T.R.; Writing – review & editing: C.A.B., W.R., K.E.D., T.R., P.M., L.J.B.

### Funding

The flume experiments were supported by funding from the Ian Potter Foundation and the NSW Recreational Fishing Trusts. Manuscript preparation and open access publication were supported through funding from the New South Wales Government and the Fisheries Research and Development Corporation, on behalf of the Australian Government, via project 2022-003, "Evaluating the economic and environmental return on investment of modern fish screens". Open Access funding provided by Fisheries Research and Development Corporation. Deposited in PMC for immediate release.

### Data and resource availability

All relevant data and details of resources can be found in the article and its supplementary information. Data are also available online at Dryad (https://doi.org/10.5061/dryad.gb5mkkx3d).

### Peer review history

The peer review history is available online at https://journals.biologists.com/bio/lookup/doi/10.1242/bio.062262.reviewer-comments.pdf

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
