## [Peer Review File · Biology Open]

Protecting larval fish at water intakes: Hydraulic and biological evidence for the effectiveness of modern fish-protection screens

Wayne Robinson, Katherine E. Doyle, Tom Rayner, Patrick McSweeney, Lee J. Baumgartner and Craig Ashley Boys

DOI: 10.1242/bio.062262

Editor: Lewis Halsey

Review timeline

Original submission: 25 September 2025

Editorial decision: 1 October 2025

First revision received: 2 October 2025

Accepted: 2 October 2025

Original submission

First decision letter

MS ID#: bio.062262

MS Title: Protecting larval fish at water intakes: Hydraulic and biological evidence for the effectiveness of modern fish-protection screens

Authors: Craig Ashley Boys; Wayne Robinson; Katherine E. Doyle; Tom Rayner; Patrick McSweeney; Lee J. Baumgartner

Dear Dr Boys,

I have now reached a decision on the above manuscript.

The reviewer reports are shown at the bottom of this email or can be accessed, together with a copy of this decision letter, by going to:

.

As you will see, the reviewers gave favourable reports, but raised some minor critical points that will require amendments to your manuscript. I hope that you will be able to carry these out, because we would like to be able to accept your paper.

Reviewer 1

Comments to Author

Overall Summary: This manuscript is an effort towards addressing whether entrainment criteria for fish species currently used in Australia are enough to protect juvenile fish species as well. They found that approach velocities and slot size of protective mesh used for this purpose had significant and interacting effects, all together, on larval entrainment (entrapment). However, they also provide evidence that hydraulics are the most important and driving factor in protectiveness (more so than slot size of the mesh). This is solid work, that resulted in validation of current criteria also for juveniles, and warrants a revision in criteria used in countries around the world.

I have a few minor comments, and line by line suggestions, that could be used or addressed to improve the manuscript:

54: synthesiSed to synthesized

57: revised "multi-species" typos

83: why the Murray cod? I suggest discussing the advantage or the reason why this is the ideal model species to test your parameters

107: What is VelY? Unclear at this point

114: here it is clear (Sweeping Velocity), but VelY was not clear when first introduced

Comment: The size of the juvenile Murray cod is unclear (the exact size or ranges)

Comment: Lines 310 - 325 can be re-written or merged with the conclusion paragraph. Both are written well, but as I read the conclusion, it felt redundant with these previous lines. Just a suggestion.

350 - 360 - This paragraph should be in the introduction instead of Methods section. It is useful information when introducing the model used in the study.

363 - Fish collected in this work were from 6 years ago? So these experiments were performed 6 years ago? Is this the only batch? Could there be batch-effects that could show variability from one batch to another? It is not clear if data is from the same spawning event.

Fig 5 - I suggest designing a simplified schematic that could describe the experimental design. I also suggest using the experimental design (which could be a new figure) earlier in the manuscript, and not being the final figure. It is needed to visually understand the entrainment/ no entrainment results.

Reviewer 2

Comments to Author

This strikes me a very professional and thorough study with clear conclusions of significant practical value for protecting river fish populations from being depleted by human infrastructures. Moreover, their hydrodynamic measurements help explaining the results and make the study quite mechanistic and therefore relevant from a basic biological / biomechanical perspective.

The term "entrainment" is frequently used in the manuscript. Many may not be familiar with its meaning. When I googled it, it became clear that it is a term used in many areas with different meanings. It would be good if its meaning in the current context is explained when first mentioned.

Minor comments

Line 57: "multi-species" misspelled.

Line 176: It is stated that "Proximity to the screen strongly influenced entrainment in unscreened treatments" is a bit odd since you cannot be close to something that is not there. Maybe it can be better formulated like "Proximity to the intake ..."?

Line 357: Change "day post hatch" to "day post hatch (DPH)" to properly introduce the abbreviation that is subsequently used.

Reviewer's Responses to Questions

Experimental quality

Does each figure have the proper controls?

If 'No', please indicate reasons in Comments for Author box below.

Reviewer #1:

- Yes

Reviewer #2:

- Yes

Were the data analyzed using appropriate statistical tests?

If 'No', please indicate reasons in Comments for Author box below.

Reviewer #1:

- Yes

Reviewer #2:

- Yes

Reproducibility

Were experiments performed using adequate number of biological replicates?

If 'No', please indicate reasons in Comments for Author box below.

Reviewer #1:

- Yes

Reviewer #2:

- Yes

Does the methods section provide sufficient detail to permit reproducibility?

If 'No', please indicate reasons in Comments for Author box below.

Reviewer #1:

- Yes

Reviewer #2:

- Yes

Completeness

Are the manuscript's conclusions supported by the data?

If 'No', please indicate reasons in Comments for Author box below.

Reviewer #1:

- Yes

Reviewer #2:

- Yes

Scholarship

Do the authors cite and discuss the merits of data that would argue for and against their conclusion?

If 'No', please indicate reasons in Comments for Author box below.

Reviewer #1:

- Yes

Reviewer #2:

- Yes

Does the manuscript title & abstract accurately reflect the contents of the manuscript, without hyperbole?

If 'No', please indicate reasons in Comments for Author box below.

Reviewer #1:

- Yes

Reviewer #2:

- Yes

First revision

Responses for handling editor.

Hello, and thank you for the rapid turnaround.

I've incorporated the reviewer's suggestions wherever possible. Where changes have not been made, these are explained in the responses below.

In addition to addressing reviewer comments, I note that the journal allows a maximum of eight display items and has requested that one be removed. In response, I have removed the original Figure 4 (the photo montage).

Reviewer 1:

Overall Summary: This manuscript is an effort towards addressing whether entrainment criteria for fish species currently used in Australia are enough to protect juvenile fish species as well. They

found that approach velocities and slot size of protective mesh used for this purpose had significant and interacting effects, all together, on larval entrainment (entrapment). However, they also provide evidence that hydraulics are the most important and driving factor in protectiveness (more so than slot size of the mesh). This is solid work, that resulted in validation of current criteria also for juveniles, and warrants a revision in criteria used in countries around the world.

I have a few minor comments, and line by line suggestions, that could be used or addressed to improve the manuscript:

54: synthesiSed to synthesized

No change made at this stage. I have adopted UK spelling throughout (i.e. synthesised over synthesized) and my understanding is that the journal does not have a preference for UK or US spelling, only that it is consistently applied throughout the manuscript. Please correct me if I am mistaken.

57: revised "multi-species" typos

Revised line 58

83: why the Murray cod? I suggest discussing the advantage or the reason why this is the ideal model species to test your parameters

To deal with this comment and one made below, I have moved the description of why Murray cod were used from the methods section and have incorporated it into the introduction as requested. The rewritten text is highlighted from line 83-104.

107: What is VelY? Unclear at this point

VelY has been defined previously in the Methods, but I understand that because Biology Open is structured where the Methods come at the end of the paper, some readers may jump straight to the results. I have tried to resolve this comment from the reviewer by repeating the definition of VelY in the results. See line 120 addition of "VelY, represented the component perpendicular to and directed toward the screen face" .

114: here it is clear (Sweeping Velocity), but VelY was not clear when first introduced

This has been addressed. See my response to previous comment

Comment: The size of the juvenile Murray cod is unclear (the exact size or ranges)

I'm not sure how to resolve this as I have already defined the size of this fish used in the methods section (line 355-357) - "*For the duration of the experiment these individuals measured (mean \pm S.E.): total length 13.2 ± 0.3 mm, dorso-ventral depth 2.9 ± 0.1 mm, and maximum width 2.1 ± 0.0 mm.*" please let me know if you think the reviewer is making a point that I've missed.

Comment: Lines 310 - 325 can be re-written or merged with the conclusion paragraph. Both are written well, but as I read the conclusion, it felt redundant with these previous lines. Just a suggestion.

This is a good suggestion as there was a fair bit of replication between these adjacent paragraphs. I've addressed this by combining the two sections as suggested. See lines 316-338

350 - 360 - This paragraph should be in the introduction instead of Methods section. It is useful information when introducing the model used in the study.

Agreed, this has been deleted and now incorporated in the introduction. See lines 83-104

363 - Fish collected in this work were from 6 years ago? So these experiments were performed 6 years ago? Is this the only batch? Could there be batch-effects that could show variability from one batch to another? It is not clear if data is from the same spawning event.

Correct the experiments were 6 years ago as a single batch therefore there are no issues around batch effect. Because of this and to remove the chance of unnecessary confusion I have removed reference to the year. See line 342-343.

Fig 5 - I suggest designing a simplified schematic that could describe the experimental design. I also suggest using the experimental design (which could be a new figure) earlier in the manuscript, and not being the final figure. It is needed to visually understand the entrainment/ no entrainment results.

While I appreciate the reviewers suggestion, due to the journal's limit of eight figures/tables, I'm unable to include an additional schematic without removing existing figure or table. While I agree that clearly articulating the experimental design is important, this is already presented in Table 4 within the methods. Although Biology Open places the results before the methods, I expect readers seeking context for the results will refer to the experimental design table at that point.

To address the concern about clarity, I've expanded the caption of Figure 5 to more clearly explain how the no-screen treatment differs from the screened treatment (see highlighted changes). I considered replacing the table with a schematic figure, but I believe the current table presents the experimental treatments more clearly and completely than a simplified diagram would allow.

Reviewer 2:

This strikes me a very professional and thorough study with clear conclusions of significant practical value for protecting river fish populations from being depleted by human infrastructures. Moreover, their hydrodynamic measurements help explaining the results and make the study quite mechanistic and therefore relevant from a basic biological / biomechanical perspective.

The term "entrainment" is frequently used in the manuscript. Many may not be familiar with its meaning. When I googled it, it became clear that it is a term used in many areas with different meanings. It would be good if its meaning in the current context is explained when first mentioned.

I've added a short definition to paragraph one of the introduction line 31. I've also added a more detailed definition of entertainment to the methods section line 481-483.

Minor comments

Line 57: "multi-species" misspelled.

Corrected line 58.

Line 176: It is stated that "Proximity to the screen strongly influenced entrainment in unscreened treatments" is a bit odd since you cannot be close to something that is not there. Maybe it can be better formulated like "Proximity to the intake ..."?

Corrected, now reads proximity to the intake (line 190).

Line 357: Change "day post hatch" to "day post hatch (DPH)" to properly introduce the abbreviation that is subsequently used.

Corrected, abbreviation now used with full term at first mention (line 93).

Second decision letter

MS ID#: bio.062262R1

MS Title: Protecting larval fish at water intakes: Hydraulic and biological evidence for the effectiveness of modern fish-protection screens

Authors: Craig Ashley Boys; Wayne Robinson; Katherine E. Doyle; Tom Rayner; Patrick McSweeney; Lee J. Baumgartner

Dear Dr Boys,

This morning I've read thoroughly through your Reviewer responses. I am happy to tell you that I am accepting your manuscript for publication in Biology Open, pending our standard publication integrity checks. It was accepted on 02 October 2025.